# Blood Glucose Levels during Decathlon Competition: An Observational Study in Timing of Intake and Competing Time

**DOI:** 10.3390/metabo14010047

**Published:** 2024-01-12

**Authors:** Rikako Yoshitake, Hitomi Ogata, Naomi Omi

**Affiliations:** 1Graduate School of Comprehensive Human Sciences, University of Tsukuba, 1-1-1 Tennodai, Tsukuba 305-8574, Japan; s2230490@s.tsukuba.ac.jp; 2Graduate School of Humanities and Social Sciences, Hiroshima University, 1-7-1 Kagamiyama, Higashi-Hiroshima 739-8521, Japan; hogata@hiroshima-u.ac.jp; 3Institute of Health and Sport Sciences, University of Tsukuba, 1-1-1 Tennodai, Tsukuba 305-8574, Japan

**Keywords:** blood glucose levels, flash glucose monitoring, decathlon, performance, intake

## Abstract

During a men’s decathlon, a combined event conducted over two consecutive days, fluctuations in blood glucose were measured using flash glucose monitoring. Because decathletes repeatedly intake and exercise, high and low blood glucose levels are observed, but the actual conditions have not yet been clarified. Low blood glucose levels (<80 mg/dL) were observed in nine athletes, while high blood glucose levels (>139 mg/dL) were observed in all athletes at least once during the competition days. Furthermore, low blood glucose levels were observed in nine athletes at least once during and after intake (“intake” refers to consuming energy-containing food and beverages). Additionally, high blood glucose levels were observed in nine athletes at least once during and after intake. Five athletes had low blood glucose during competing time. It was suggested that even if they had eaten a meal just prior to the competition, their intake was likely insufficient for their energy expenditure. A significant positive correlation was found between the mean blood glucose level and the number of intakes on competition days. It is believed that meals may have had a strong influence on blood glucose, even on competition days with a high frequency of eating and exercise for the decathlon.

## 1. Introduction

Combined events are one of the events for track and field, consisting of the men’s decathlon and the women’s heptathlon. In the men’s decathlon, events unfold over two consecutive days, featuring the 100 m sprint, long jump (LJ), shot put (SP), high jump (HJ), and 400 m sprint on the first day, followed by the 110 m hurdles (110 mH), discus throw (DT), pole vault (PV), javelin throw (JT), and 1500 m race on the second day. The points scored for each event are scored based on a predefined scoring table, and the winners are determined by the sum of their results. From a physiological perspective, all these events, excluding the 1500 m, are primarily considered anaerobic exercises, as they involve short periods of running, jumping, and throwing [1]. Conversely, in national-level and world-class decathletes, V∙O2MAX values of 52.1 and 56.7 mL/kg/min, respectively [2,3], have been reported, indicating the presence of anaerobic and endurance components.

HJ, LJ, JT, and PV are 6 METs and hurdles are 10 METs, and the speed shown in the sprint events is much faster than 23 METs (22.5 km/h) [4]. Therefore, a decathlon, which includes more than half of these events, is a high-intensity exercise regimen. According to the International Olympic Committee, maintaining adequate nutrition is necessary to achieve high performance [5]. Given the intense nature of decathlon events and the prolonged duration of the competition, the impact of nutrition is speculated to be especially significant within this discipline [6]. One essential indicator of nutritional status on competition days is blood glucose levels, as maintaining them within normal limits is vital for optimal physiological function and performance during exercise [7]. Notably, continuous monitoring of blood glucose levels has been undertaken in marathon and cycling events to provide insights into effective nutritional strategies during competitions [8,9,10]. In athletes, specifically jumping and throwing athletes, pre-competition diets should aim to maintain satiety and stable blood glucose levels throughout the competition [6]. Regular physical activity typically enhances insulin sensitivity [11], but high-intensity exercise causes rapid consumption of glucose in the blood and muscles, leading to a low blood glucose level, but it is thought that the blood glucose level may rise temporarily as a result of the liver’s response to maintain constant blood glucose levels. During the competition days, both low blood glucose levels due to insufficient intake and high blood glucose levels resulting from intense exercise may manifest. However, the actual conditions have not yet been clarified. Furthermore, the rapid cycle of intake and high-intensity exercise makes blood glucose management challenging, necessitating a comprehensive understanding of blood glucose levels to develop effective intake strategies. Recently, the use of flash glucose monitoring (FGM) devices, allowing for noninvasive and calibration-free daily glucose profiling, has become increasingly prevalent. Although the FGM thresholds in athletes remain unclear, conventional guidelines suggest that the normal range is between 80 and 139 mg/dL, with levels above 140 mg/dL considered high and levels below 80 mg/dL considered low [12]. Therefore, this study aims to assess blood glucose levels during the competition days of decathletes. Additionally, it seeks to explore the effects of the timing of intakes and competing periods on the occurrence of high and low blood glucose levels. Throughout this study, the two competition days of the decathlon will be referred to as “competition days”, while the periods encompassing each event, including warm-up, will be denoted as “competing time”. 

## 2. Materials and Methods

All subjects gave their informed consent for inclusion before they participated in this study. The study was conducted in accordance with the Declaration of Helsinki, and the protocol was approved by the Ethics Committee of the University of Tsukuba (No. tai 020-51).

### 2.1. Participants

The subjects comprised 12 healthy university or graduate students who specialized in decathlon, ranging from beginner- to national-level competitors. Among them, two athletes who entered but did not participate in the competition were excluded from the study. The final sample comprised 10 athletes who participated in the official decathlon competition between July and October 2022 and competed over the two days. Two of these 10 athletes could not complete in one event out of the ten events. The mean age of the athletes was 20.2 ± 1.1 years. Detailed information about the athletes is shown in Table 1.

### 2.2. Experimental Design and Measurements

The study was conducted over 4 d, encompassing the two decathlon competition days and one day before and after the competition. It included continuous blood glucose monitoring, examination of intake timing, activity tracking, and video recording. Diurnal fluctuations in blood glucose levels were recorded using a continuous glucose monitoring system (Freestyle Libre Pro, Abbott, Chicago, IL, USA). The sensor was inserted under the skin on the back of the upper arm and levels of interstitial glucose concentrations were measured. The measurements were then wirelessly transmitted from the sensor to the receiver. The monitoring device collects average data at 15 min intervals and can record blood glucose levels continuously over a 24 h period for up to 2 weeks. Notably, the device has a mean absolute relative difference of 11.4% [13] and does not require calibration. To establish precise exercise durations, the athletes maintained a comprehensive activity log, and to ascertain the ends of their competing times, we recorded the athletes’ performances using a video camera. Individual intake times were reported immediately before intake. Height was measured using a height meter (NAVIS, YS-OS, Tokyo, Japan), and weight was measured using bioelectrical impedance analysis (TANITA, MC-190, Tokyo, Japan). Additionally, body fat percentage was determined using dual-energy X-ray absorptiometry (Hologic Horizon, Marlborough, MA, USA: DXA). These measurements were taken during the competition season in the same year on the day of measurement. The experimental protocol is shown in Figure 1.

### 2.3. Data Analysis

The analysis interval was 48 h from the wake-up time on the first day of the decathlon. As the obtained blood glucose levels may not be stable within 24 h of attachment, we used data from sensors attached over 24 h earlier, excluding cases where the sensors were dislodged during the measurement period, such as ID: 3 and ID: 12. Additionally, ID: 1 (Day 1: 7:15–7:30; Day 2: 17:45–18:00, 18:15–18:30, 20:30–6:00), ID: 3 (Day 1: 5:00–9:00), ID: 6 (Day 2: 23:30–7:00), and ID: 10 (Day 1: 8:00–14:15), where sensor dislodgment or missing data occurred, were excluded from the calculation. In addition, the coefficient of variation (CV; SD/average blood glucose level × 100), which is the main blood glucose fluctuation index in the international consensus report, was calculated [14] Although there are individual differences in blood glucose levels, the subjects (*n* = 10) analyzed in this study were within normal values (CV < 36%). The competing times for each event were based on the start time of the warm-up and were calculated using the athletes’ activity logs. The ends of competing times were calculated based on individual video recordings. In this study, low blood glucose levels were defined as below 80 mg/dL, while high blood glucose levels were characterized as above 139 mg/dL [13]. The blood glucose levels presented in this study were recorded by counting the occurrences of low or high blood glucose as one epoch of data, with the mean values calculated every 15 min. Furthermore, CGMS exhibits a physiological time delay in acquiring interstitial fluid glucose values with respect to changes in blood glucose levels [15]. In this study, considering the short-term measurement of 2 days and the targeting of young, healthy athletes, the start time of the data acquired once every 15 min was adjusted to compensate for physiological delays (e.g., 8:07–8:22 → 8:00–8:15). We categorized occurrences of high and low blood glucose levels appearing within 2 h in each respective epoch into five categories: competing time, after competing time, before breakfast (from waking up until breakfast), IT (during and after intake), and sleep. Incidents not fitting these categories were classified as “etc.”, resulting in a total of six categories. IT (intake) refers to the consumption of any food and beverages containing energy, regardless of type or quantity. Furthermore, instances of low or high blood glucose induced by intake during competing times were categorized as IT. When low or high blood glucose levels persisted across consecutive epochs during competing times, the initial epoch was categorized as IT, while the subsequent two or more epochs were classified as competing time. Multiple intakes occurring within 15 min were treated as a single intake. Details are shown in Figure 2.

### 2.4. Statistical Analysis

All data are presented as mean ± standard deviation. Kolmogorov–Smirnov test was used to test the normality of the data, and they were normally distributed. Pearson’s correlation analyses were conducted to explore the relationship between the average blood glucose levels over the 2 days and the number of intakes. All statistical analyses were conducted using SPSS (version 29; IBM Inc., Chicago, IL, USA). The significance level was set at α = 0.05.

## 3. Results

### 3.1. Changes in Blood Glucose Levels in the Decathlon

Figure 3 shows fluctuations in blood glucose levels for ID: 5, where no data were missing. The longer and lighter-colored vertical areas indicate competing times, while the longer and darker vertical areas indicate sleeping times. Long horizontal colored areas indicate blood glucose levels within the normal limits defined in this study (80–139 mg/dL). The arrows indicate the intake of all food and beverages containing energy, and light food was taken as appropriate during the short rest periods. On competition days, the mean blood glucose level for ID: 5 was 99.3 ± 18.4 mg/dL, ranking as the fourth lowest among the 10 athletes. Moreover, ID: 5 had the third lowest number of intakes, totaling 16.

### 3.2. Blood Glucose Levels and Number of Intakes and Cumulative Distribution Plots in the Decathlon

Table 2 shows details of blood glucose levels during the competition days. Fasting blood glucose was defined as blood glucose levels within 15 min of waking up on Day 1 of the decathlon. The average value for the measurement period (48 h) was 104.1 mg/dL, with a mean maximum value of 162.8 mg/dL and a mean minimum value of 67.4 mg/dL. Individually, ID: 6 exhibited the highest mean blood glucose level (124.3 mg/dL), while ID: 9 exhibited the lowest mean blood glucose level (87.8 mg/dL). For reference, the highest and lowest blood glucose values were 206 mg/dL (ID: 6) and 40 mg/dL (ID: 9), with 40 mg/dL being the lowest measuring value of the device. The average number of intakes was 19 over the two days, with 10.1 on the first day and 9.1 on the second.

Figure 4 depicts the extent to which blood glucose levels fell outside the normal range (80–139 mg/dL) on competition days. Over half of the athletes spent more than 10% of their time with blood glucose levels outside the normal range of 80–139 mg/dL during the competition days (48 h).

### 3.3. Details of High and Low Blood Glucose Occurrence

Table 3 presents details of the number of times when values above and below the normal range of blood glucose levels (80–139 mg/dL) were observed during the competition days. During the competition days, low blood glucose was observed at least once in nine out of ten athletes, while high blood glucose was observed in all athletes. These occurrences were classified into six categories: competing time, after competing time, before breakfast (from waking up until breakfast), IT (during and after intake), sleep (sleeping time), and “etc.” within 2 h of the epoch in which low or high blood glucose was observed. Nine of the ten athletes exhibited both low and high blood glucose levels on IT. During the competing time, high blood glucose was observed in eight out of the ten athletes, and low blood glucose was observed in five out of the ten athletes. After competing time, six out of the ten athletes had low blood glucose levels, while three out of the ten athletes had high blood glucose levels. Furthermore, during sleep, six out of the ten athletes exhibited low blood glucose levels, and before breakfast, four out of the ten athletes had low blood glucose levels. Notably, none of the athletes exhibited high blood glucose levels before breakfast or during sleep. Overall, high and low blood glucose levels were commonly observed during IT rather than in the competing time or during sleep.

### 3.4. Cumulative Distribution of Blood Glucose Levels during the Decathlon

Figure 5 shows the relationship between mean blood glucose level and the number of intakes. A significant positive correlation was observed between mean blood glucose levels and the number of intakes over the two days (r = 0.713, *p* < 0.05).

## 4. Discussion

This observational study is the first attempt to continuously monitor blood glucose levels during a decathlon competition. The results reveal that both low and high blood glucose levels occur during decathlon competitions. While there were individual differences, it became evident that repeated cycles of intake and exercise led to fluctuations in blood glucose levels, with frequent increases and decreases. In this study, the normal range of values measured by FGM was defined as 80–139 mg/dL, with values below 80 mg/dL defined as low blood glucose and values above 140 mg/dL as high blood glucose.

Nine participants had low blood glucose, which appeared within 2 h of IT, five during the competing time, and six after the competing time. This implies that low blood glucose appeared more frequently in athletes during and after intake than during and after competing times. Furthermore, it suggests a risk of exercise-induced low blood glucose when carbohydrates are consumed 30–45 min before exercise [16,17,18]. This can lead to reduced lipolysis and increased carbohydrate utilization, which may reduce endurance exercise performance [19]. In this study, we did not observe any effect on performance. However, exercise-induced low blood glucose can occur during a decathlon, and intake strategies should be adapted to the day’s competition schedule. Consequently, risk management to prevent low blood glucose caused by both insufficient intake and the timing of intake during short intervals between competing times should be considered.

Five athletes were observed to have low blood glucose during the competing time. It is implied that even if they had consumed food immediately before the event, their intake was likely insufficient in relation to their energy expenditure. Four of these athletes were either competing in their first decathlon or could not complete the decathlon. Therefore, it was suggested that it is necessary to prevent low blood glucose, especially during competing times. Epochs above 140 mg/dL were observed in nine athletes during the IT, eight during the competing time, and five after the competing time. Three athletes achieved new personal records on the day of the study. Two athletes had high 48 h mean blood glucose levels within the normal range, and one athlete was observed more often than not to be under 80 mg/dL, indicating individual differences.

Furthermore, the decathlon holds the highest dropout rate compared to other athletic events, with 22% of high-level athletes reportedly dropping out [20]. Moreover, the risks associated with the decathlon are diverse in terms of physical, psychological, and technical aspects [21,22,23]. Prolonged periods of low blood glucose levels have been linked to negative mood swings, reduced cognitive performance, and low-energy states [24]. Although the two athletes who could not complete in the decathlon in this study cited lower limb injuries as the cause for withdrawal, both athletes only had low blood glucose levels. These findings underscore the necessity of maintaining blood glucose levels for sustained high performance in the decathlon until the last event. Additionally, it can be assumed that both the action of glucagon [25,26] and the action of adrenaline associated with sympathetic activation [27], such as tension, increased the blood glucose levels. Although we did not take real-time measurements as the competition was in progress, it can be inferred that the action of both glucagon and adrenaline influenced blood glucose levels.

Although other factors may be involved, highlighting the number of intakes on competition days in the decathlon, a higher number of intakes correlated with higher average blood glucose levels. Therefore, it was considered important to ensure the frequency of intake, even during the short rest periods between competitions.

In recent years, CGMS has unveiled the potential to monitor real-time blood glucose fluctuations [28]. However, this review also highlighted several issues, such as the difficulty of conducting investigations due to match regulations and the sensor becoming detached due to sweat. Decathlon athletes also encounter challenges in self-monitoring during competitions due to the imposed regulations. Even when coaches are monitoring, maintaining proximity to athletes within the athletic track is challenging. Thus, having prior knowledge of the conditions that may lead to high or low blood glucose levels can be instrumental in devising effective dietary strategies during competitions.

This study has several limitations. First, there is a physiological time lag between changes in interstitial fluid glucose levels and blood glucose levels [15], which could introduce minor errors in our measurements. Second, our study focused on the timing of intake and did not examine the quantity and composition of intake. Therefore, future studies must clarify fluctuations in blood glucose levels, specifically focusing on the quantity and composition of the competition day intake. Third, the FGM measurements ranged from 40 to 500 mg/dL, with 40 being the lowest value recorded in our study. Since the measurements were made using FGM, we focused on the occurrence of low blood glucose over time and did not categorize them as missing values. However, future studies should assess the validity of measurements under extreme low blood glucose, such as 40 mg/dL. Fourth, in this study, we were only able to measure 10 subjects during one competition, so it was difficult to evaluate in detail the physiological relationship between abnormal blood glucose levels and individual performance. In the future, it will be necessary to conduct detailed analyses based on individual performance levels by repeatedly conducting tests on the same athletes in different competitions.

## 5. Conclusions

In conclusion, this observational study provided insight into the fluctuations in blood glucose levels during men’s decathlon competition days. A significant positive correlation between the mean blood glucose level and the number of intakes on competition days suggests meals strongly influence blood glucose levels in decathlon competitions.

## Figures and Tables

**Figure 1 metabolites-14-00047-f001:**
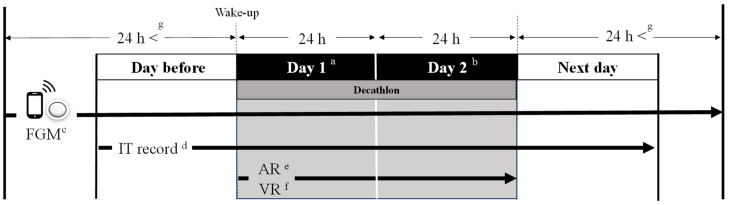
Experimental protocol. ^a^ Day 1 starts at wake-up time and ends 24 h later. ^b^ Day 2 follows Day 1 and spans another 24 h period. The FGM sensor was attached at least 24 h before the start of Day 1 and removed at least 24 h after the end of Day 2. ^c^ FGM (
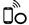
) = flash glucose monitoring. ^d^ IT record = intake record. ^e^ AR = activity record. ^f^ VR = video camera record. ^g^ The FGM sensor remained attached from at least 24 h before the start of the competition days to at least 24 h after they had ended. *n* = 10.

**Figure 2 metabolites-14-00047-f002:**
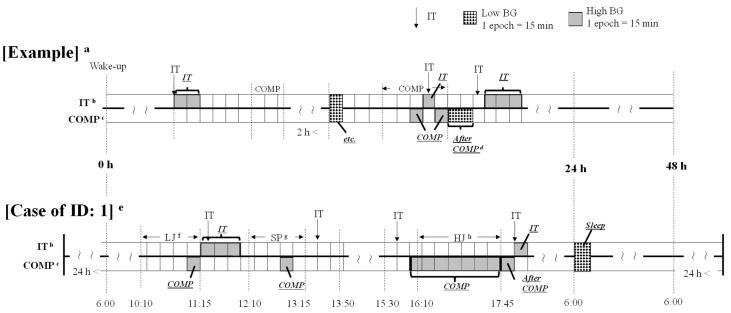
Analysis of blood glucose level epochs at the onset of high and low blood glucose levels. ^a^ Epoch categorization method for the occurrence of low or high blood glucose. ^b^ IT = intake: epoch categorization method for low or high blood glucose occurrences during intake. ^c^ COMP = competing time (running, jumping, or throwing): epoch categorization during low or high blood glucose epochs while competing. ^d^ After COMP = after competing time: epoch categorization of low or high blood glucose levels after competing time. ^e^ Results for ID: 1 (10:10–18:15) categorized based on the example method are presented in the figure. In this case, high blood glucose at 11:00 AM was categorized as COMP because LJ took place from 10:10 to 11:15 (similarly categorized for the 12:45–13:00 occurrence during SP). The subsequent three epochs were categorized as IT, as an intake occurred at 11:20. High blood glucose was observed consecutively from 16:00 to 18:15, but HJ took place from 16:10 to 17:45, resulting in all epochs from 16:00 to 17:45 being categorized as COMP and 17:45 to 18:00 being categorized as After COMP. The 18:00–18:15 interval was categorized as IT, as a meal was consumed at 18:00. Low blood glucose at 6:00 was classified as sleep because the athlete was asleep. ^f^ LJ = long jump. ^g^ SP = shot put. ^h^ HJ = high jump. *n* = 10.

**Figure 3 metabolites-14-00047-f003:**
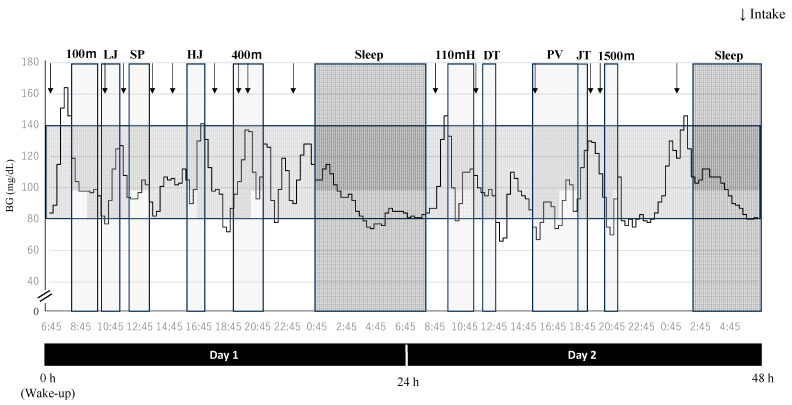
Blood glucose fluctuations during the decathlon (ID: 5). BG = blood glucose levels. LJ = long jump, SP = shot put, HJ = high jump, 110 mH = 110 m hurdle, DT = discus throw, PV = pole vault, JT = javelin throw. The wake-up time for ID: 5 was 6:45, and the sleep time between competitions was 0:30–8:00.

**Figure 4 metabolites-14-00047-f004:**
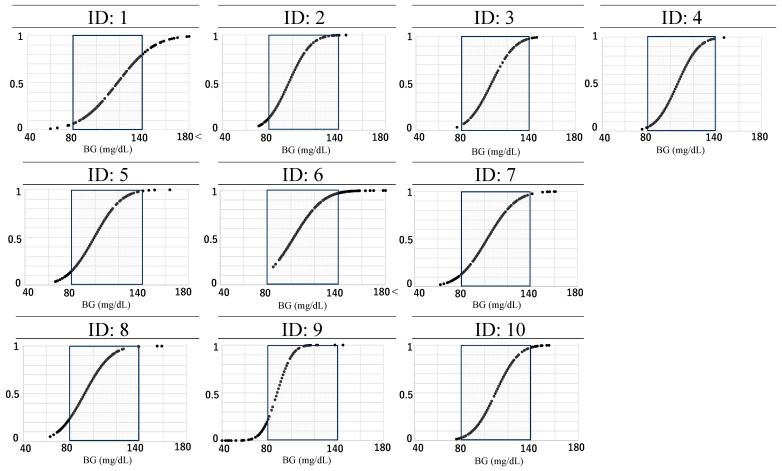
Cumulative distribution plots of measured CGM values during the decathlon competition days. BG = blood glucose levels 48 h from the wake-up time on Day 1; missing data were recorded for ID: 1 (Day 1: 7:15–7:30; Day 2: 17:45–18:00, 18:15–18:30, 20:30–6:00), ID: 3 (Day 1: 5:00–9:00), ID: 6 (Day 2: 23:30–7:00), and ID: 10 (Day 1: 8:00–14:15); *n* = 10.

**Figure 5 metabolites-14-00047-f005:**
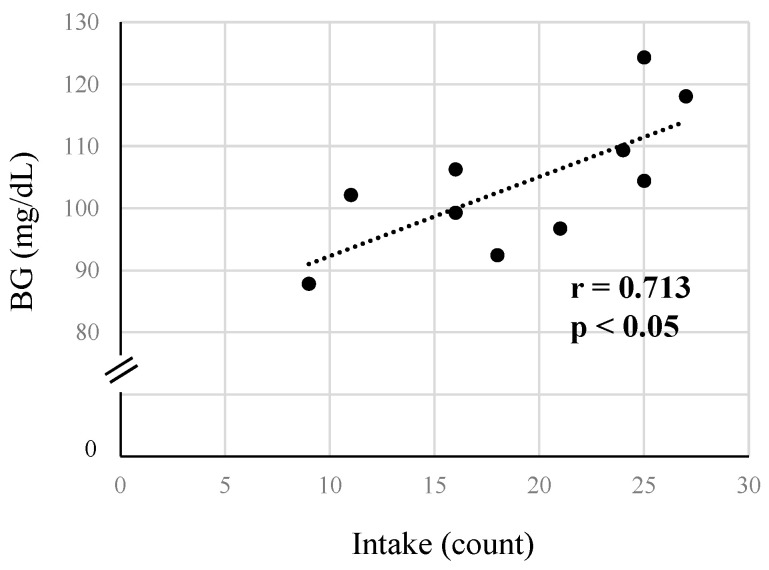
Relationship between the number of intakes and mean blood glucose levels. BG = blood glucose levels. r = Pearson product–moment correlation coefficient., *p* < 0.05; *n* = 10.

**Table 1 metabolites-14-00047-t001:** Athletes’ anthropometric characteristics, completed events, and results.

ID	Age	Height (cm)	Weight (kg)	BMI (kg/m^2^)	Fat Percentage (%)	Running ^a^ (4 Events)	Jumping ^b^ (3 Events)	Throwing ^c^ (3 Events)	WAS ^d^	Results
1	21	184.2	74.0	21.8	10.7	○	○	○	928	PB
2	20	177.5	69.9	22.2	16.6	3/4 ^e^	○	○	417	PB *
3	19	183.2	82.8	24.7	15.5	○	○	○	801	PB *
4	22	173.3	69.5	23.1	9.9	3/4 ^e^	○	○	887	
5	21	180.6	79.7	24.4	15.2	○	○	○	873	PB
6	21	181.9	73.3	22.2	13.1	○	○	○	879	PB
7	21	183.2	85.5	25.5	14.9	○	○	○	852	
8	19	171.3	64.8	22.1	14.6	○	○	○	699	PB *
9	19	165.5	65.2	23.8	11.6	○	○	○	805	PB *
10	19	180.2	72.3	22.3	12.7	○	○	○	628	PB *
Mean ± SD	20.2 ± 1.1	178.1 ± 6.2	73.7 ± 7.0	23.2 ± 1.3	13.5 ± 2.2					

PB = personal best. PB * = personal best (first decathlon). ^a^ 100 m, 400 m, 110 mH, 1500 m; ^b^ LJ, HJ, PV, ^c^ SP, DT, JT. ^d^ WAS = WORLD ATHLETICS SCORING TABLES OF ATHLETICS. ^e^ Participated in three of the four events and skipped one due to a lower limb injury. *n* = 10.

**Table 2 metabolites-14-00047-t002:** Blood glucose level information.

	Blood Glucose Level (mg/dL)						Intake (Count)		
ID	FBG ^a^	Average	SD	CV	Max	Min	Day 1	Day 2	2 Days
1	99	118.0	25.3	21.5	199.0	60.0	15	12	27
2	73	96.7	15.2	15.7	146.0	71.0	9	12	21
3	- ^b^	106.2	16.4	15.5	146.0	76.0	8	8	16
4	85	104.4	15.0	14.3	147.0	75.0	13	12	25
5	89	99.3	18.4	18.6	164.0	66.0	10	6	16
6	116	124.3	22.2	17.9	206.0	85.0	12	13	25
7	68	102.1	19.8	19.4	161.0	62.0	7	4	11
8	73	92.4	16.0	17.3	159.0	63.0	11	7	18
9	78	87.8	15.8	18.0	145.0	40.0 ^c^	5	4	9
10	83	109.3	17.6	16.1	155.0	76.0	11	13	24
Average	84.9	104.1	18.2	17.4	162.8	67.4	10.1	9.1	19.2

^a^ FBG = fasting blood glucose. ^b^ No data for ID: 3 due to missing data. ^c^ It was 40 in two consecutive epochs at 3:15–3:45. Missing data: ID: 1 (Day 1: 7:15–7:30; Day 2: 17:45–18:00, 18:15–18:30, 20:30–6:00), ID: 3 (Day 1: 5:00–9:00), ID: 6 (Day 2: 23:30–7:00), and ID: 10 (Day 1: 8:00–14:15). *n* = 10.

**Table 3 metabolites-14-00047-t003:** Details of the occurrence of high (>139 mg/dL) and low (<80 mg/dL) blood glucose levels.

	>139 mg/dL						<80 mg/dL					
ID	Before BF ^a^	IT ^b^	COMP ^c^	After COMP	Sleep	etc.	Before BF ^a^	IT ^b^	COMP ^c^	After COMP	Sleep	etc.
1		9	13	3				3			3	
2		1	1				1	5	1	1	17	
3		3	1	2				1				
4			1					3				
5		5	1					4	8	8	6	
6		15	14	8		1						
7		2	1	3			3	3	1	5	9	2
8		2					7	5	6	8	16	2
9		1					3	6	2	11	28	3
10		10	3	3				1		1		
People ^d^	0	9	8	5	0	1	4	9	5	6	6	3

^a^ Before BF = before breakfast. ^b^ IT = intake. ^c^ COMP = competing time. ^d^ People = total number of athletes, *n* = 10.

## Data Availability

The data presented in this study are available in the main article.

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
