# Peer review of "Blood Glucose Levels during Decathlon Competition: An Observational Study in Timing of Intake and Competing Time"

_metabolites, 2024, doi:10.3390/metabo14010047_

Round 1
Reviewer 1 Report
Comments and Suggestions for Authors
Journal Metabolites (ISSN 2218-1989)
Manuscript ID metabolites-2763781
Type Article
Title Fluctuations in blood glucose levels during a decathlon competition; Focusing on the timing of intake and competing time.
As indicated from the recent review (ref 24 in the m/s) the introduction of CGM into the domain of the sports performer is a relatively new venture and studies as represented by this m/s are required to validate and guide researchers and the sports performance community as to the potential advantages and pitfalls. The submission of this study is therefore to be welcomed, but the temptation to over-analyse and to draw insufficiently substantiated conclusion(s) from what is essentially observational data should be avoided.
The rationale for this study as presented in the introduction requires further consideration that may affect the route to data analysis and outcome(s).
Comments to the authors.
1. The authors could benefit from further consideration of the recent review and guideline(s) paper on the use of CGM in sport https://doi.org/10.1123/ijsnem.2022-0139 (cited as ref 24)
2. This study is of CE athletes, but the introduction concentrates on the blood glucose levels of high endurance athletes (refs 6,8,9) with only 1 general review (ref 5) for CE athletes.
3. Supporting references to the arguments presented are either non-available (ref 11,12 see comments on references below) or unjustified, e.g., ref 11, which appears to be related to the obese for whom glucoregulation differs widely from that of the CE athlete.
4. A central tenet is the high blood glucose following ‘intense’ exercise in CE, yet no supportive evidence is provided.
5. The aim as stated is, essentially, observational (Figure 3 acts as a prime example), but the data analysis is rather oppressive in its detail, the relevance of which is not always evident to the reader.
2.3 Data analysis. A complex array of categories and epochs. For clarity, consider simplifying into a Table. Figure 2 is complex and not easily understood
Line 167/8 : Ref 14 records ‘the physiological delay of glucose transport from the vascular to the interstitial space is 5-6 min’, not 5-10 minutes as stated. In addition, the study (ref 14) was conducted using CMA microdialysis catheters inserted into the abdomen. Relevant studies related to the Abbott CGM at the upper arm include delay time https://doi.org/10.1177/1932296815590154 and ref 13 in the m/s (plus the editorial response to this paper https://doi.org/10.1089/dia.2019.0343 ) and method of averaging https://doi.org/10.1007/s12020-012-9765-1 for these devices.
2.4 Data analysis.
Line 216/7 ‘To assess fluctuations in blood glucose levels during the competition regarding the occurrence of high and low blood glucose levels, we calculated blood glucose levels throughout the competition days’
? What component(s) of the blood glucose levels was(ere) ‘calculated’
? what research question(s) and/or hypothesis(es) was(ere) tested conducting this(ese) analysis(es)
? Did these data sets qualify for parametric statistical analysis
? The use of Spearman’s rank order correlation suggests these specific data were not normally distributed
3. Results
Given the relative (in)accuracy and precision of measurement reporting values of blood glucose to 1dp is probably unwarranted. As data are presented as the mean (SD) the presumption is these data are normally distributed?
Figure 3 is the clearest representation of these data to the reader and provides the basis for consideration of the data outputs that follow.
The count of occurrences per category of event in Table 3 is somewhat misleading as it does not consider the number of occurrences per event i.e., all occurrences of low blood glucose when competing (COMP) for a subject could have occurred during a single COMP session for that subject. This is compounded when these data are summated and then averaged over the number of participants whose data contribute to this analysis. Difficult to resolve a meaningful outcome to this level of analysis. Referring to Figure 3, the magnitude (not analysed) and time for which the blood glucose deviates from the ‘normal’ range appears minimal. On that basis have the authors considered the physiological relevance of the magnitude*time deviations from the ‘norm’ to the participant’s performance? (see discussion, which appear to concentrate on endurance performance, not CE and refers to ‘prolonged events of low blood glucose’)
The cumulative distribution analysis that follows represents (to this reader) further over-analysis of these observational data.
Figure 6 is important as these analyses forms the basis of the major outcome and conclusion (line 486/6), i.e., ‘Increased the frequency of energy-containing foods and beverages is essential to prevent the risk of low blood glucose during competition days.’ The evidence in support of this conclusion is weak. The average intake frequency of the 10 subjects are parsed into 3 groups of n=5, n=2 and n=3, respectively in relation to the % of events in which low (>10%) or high (>10%) or not classified blood glucose occurred, and subjected to statistical analysis by ANOVA (??)
Do the authors consider these data are sufficient to support the conclusion offered?
References:
Unable to trace reference 11. Is this abstract/proceedings from a meeting?
Yale JF, Leiter LA, Marliss EB. Insulin resistance during recovery from strenuous exercise is greater in obesity. Intl Obes. 1985;9:A108
Unable to trace reference 12 or the stated Journal.
12. Nagata K, Shiwa G, Otsuki C, Katsuhiko K. Classification of blood glucose levels in the Flash Glucose Monitoring(FGM)era―Focusing on hypoglycemia and blood glucose level spikes―. Comprehensive Medicine.2020;19(1):21-30
Comments on the Quality of English LanguageModerate editing of English language required
Author Response
Dear Reviewer 1
We thank the referees for carefully reading our manuscript, now titled “Blood glucose levels during a decathlon competition: A descriptive study of timing for intake and competing time” and for their insightful comments. All corrections are highlighted in RED in the REVISED manuscript. Please check revised manuscuript.
Based on the referees’ comments, we have revised the manuscript and look forward to its future acceptance for publication in Metabolites.
Our responses to the referees’ comments are as follows:
Manuscript ID: metabolites-2763781
N Omi PhD
- The authors could benefit from further consideration of the recent review and guideline(s) paper on the use of CGM in sport https://doi.org/10.1123/ijsnem.2022-0139 (cited as ref 24)
Answer:
In light of all your suggestions, we can state that this study is only descriptive and have changed the title to “Blood glucose levels during a decathlon competition: A descriptive study of timing for intake and competing time”
The purpose of this study was revised to focus on the blood glucose levels.
We also added to the discussion that caution should be exercised in the use and interpretation of CGMS in athletes, and discussed how this study should be applied in the future.
(line 69) Therefore, this study aims to assess blood glucose levels during the competition days of decathletes.
(line 514) However, this review also highlighted several issues, such as the difficulty of conducting investigations due to match regulations and the sensor becoming detached due to sweat. Decathlon athletes also encounter challenges in self-monitoring during competitions due to the imposed regulations.
- This study is of CE athletes, but the introduction concentrates on the blood glucose levels of high endurance athletes (refs 6,8,9) with only 1 general review (ref 5) for CE athletes.
Answer: (line 538)
Thank you for your suggestion. Unfortunately, there is currently very little prior research on mixed competitions because it is very difficult to conduct research during a mixed competition match. Especially with regard to physiological factors, very little has been demonstrated. In fact, when searching for “decathlon” and “track and field” on PubMed, only 12 results were found (December 11, 2023). We have set this as the purpose of the discussion. However, as you pointed out, this is a descriptive paper, so we have changed the overly worded conclusion to the following.
Increasing the frequency of energy-containing foods and beverages is ideal to prevent the risk of low blood glucose during the competition day.
- Supporting references to the arguments presented are either non-available (ref 11,12 see comments on references below) or unjustified, e.g., ref 11, which appears to be related to the obese for whom glucoregulation differs widely from that of the CE athlete.
Answer: (References)
Thank you for your suggestion. Reference 11 has been removed because its citation to this paper was inappropriate. Reference 12 is a Japanese review article that describes the reference values of FGM used in this study, and we would like to cite it. Therefore, we have added "in Japanese" at the end of the citation and added the doi of this paper for your consideration. (https://doi.org/10.32183/ifcm.19.1_21)
- A central tenet is the high blood glucose following ‘intense’ exercise in CE, yet no supportive evidence is provided.
Answer: (line 40)
Yes, the referee is right, there is a paucity of literature on decathletes. According to the METs table, the high jump, long jump, javelin, and pole vault are 6 METs, the hurdle run is 10 METs, and the fastest running speed specified is 23 METs (375.4 km/h), so we also consider the competition to be high intensity because it includes more than half of these events. We have added these data to the revised text.
HJ, LJ, JT, and PV are 6 metabolic equivalents of task (METs), the hurdle is 10 METs, and the running speed is much faster than the 100 m sprint and 400 m sprint of 23 METs (375.4 km/hour) [4] suggesting that decathlon, which includes more than half of these events, is a high intensity exercise regimen.
- The aim as stated is, essentially, observational (Figure 3 acts as a prime example), but the data analysis is rather oppressive in its detail, the relevance of which is not always evident to the reader.
Answer (Figure 3)
Thank you for pointing this out. In this study, we descriptively examined blood glucose levels during the decathlon, so it is necessary to visually show how blood glucose changes, intake and exercise are repeated during the decathlon. Therefore, we would like to present the representative data first, and then describe the detailed results.
- 3 Data analysis.
A complex array of categories and epochs. For clarity, consider simplifying into a Table. Figure 2 is complex and not easily understood
Answer: (figure 2)
Thank you for pointing this out. Specific examples to the diagram were added and the font size was improved.
- Line 167/8 : Ref 14 records ‘the physiological delay of glucose transport from the vascular to the interstitial space is 5-6 min’, not 5-10 minutes as stated. In addition, the study (ref 14) was conducted using CMA microdialysis catheters inserted into the abdomen. Relevant studies related to the Abbott CGM at the upper arm include delay time https://doi.org/10.1177/1932296815590154 and ref 13 in the m/s (plus the editorial
response to this paper https://doi.org/10.1089/dia.2019.0343) and method of averaging https://doi.org/10.1007/s12020-012-9765-1 for these devices.
Answer:
Thank you for your suggestion. We apologize for the citation errors. As you pointed out, 14 references were inappropriate and have been removed. Also, thank you for kindly providing the reference.
We quoted (https://doi.org/10.1007/s12020-012-9765-1) regarding the 5-15 min physiological difference in CGM measurement time delay, and we understand that CGM has the potential to improve the error range over time, subject, and conditions. Since this study was a freestyle libre pro, the measurements were made over a short period of 2 days, the subjects were healthy young athletes, and the error may have been influenced by dehydration and temperature changes, which have not been clarified, we cited (https://doi.org/10.1177/1932296815590154), which is direct evidence from the measurement machine, for the description of the error range. In addition, recognizing the physiological delay and the error range, we analyzed the data acquired at 15-minute intervals over a 48-hour period for everyone in this study, according to the time of the forward time (e.g., 8:07-8:22 → 8:00-8:15). In light of the above, the following corrections have been made. The study limits were modified in the same way. The points to keep in mind about CGMS for athletes are discussed in the discussion and added to this section as an issue to be raised when CGMS is used for a long period of time in the future (line 514) However, this review also highlighted several issues, such as the difficulty of conducting investigations due to match regulations and the sensor becoming detached due to sweat. Decathlon athletes also encounter challenges in self-monitoring during competitions due to the imposed regulations.
(line 170) Furthermore, CGMS exhibits a physiological time delay in acquiring interstitial fluid glu-cose values with respect to changes in blood glucose levels [15]. In recent years, various studies have shown that long-term measurements on subjects such as diabetic patients and children can minimize measurement errors [15]. In this study, considering the short-term measurement of 2 days and the targeting of young, healthy athletes, the start time of the data acquired once every 15 minutes was adjusted to compensate for physiological delays (e.g., 8:07 – 8:22 → 8:00–8:15).
(line 526) First, there is a physiological time lag between changes in interstitial fluid glucose levels and blood glucose levels [15],
2.4 Data analysis. 
Line 216/7 ‘To assess fluctuations in blood glucose levels during the competition regarding the occurrence of high and low blood glucose levels, we calculated blood glucose levels throughout the competition days’
? What component(s) of the blood glucose levels was(ere) ‘calculated’
? what research question(s) and/or hypothesis(es) was(ere) tested conducting this(ese) analysis(es)
Answer: (line 225)
Thank you for your suggestion. As you pointed out, the information was missing, so the following was added. Regarding repeated measurements, since it was inappropriate to analyze 2 subjects, we divided the subjects into a group with low blood glucose levels of 10% or more and the other 5 subjects, and compared the average blood glucose levels and number of intakes. We have changed the text of 2.4 Data analysis. to the following, taking into account the other points you have mentioned.
All data are presented as mean ± standard deviation. Kolmogorov-Smirnov test was used to test the normality of the data, and they were normally distributed. Pearson's correlation analyses were conducted to explore the relationship between the average blood glucose levels over the 2 days and the number of intakes. Additionally, student's t-test was conducted to compare the mean blood glucose level and number of intakes at 48h by the percentage of blood glucose level.
- ? Did these data sets qualify for parametric statistical analysis
Answer: (line 225)
Thank you for your suggestion. As you pointed out, the information was missing, so the following was added.
All data are presented as mean ± standard deviation. Kolmogorov-Smirnov test was used to test the normality of the data, and they were normally distributed.
- ? The use of Spearman’s rank order correlation suggests these specific data were not normally distributed
Answer: (line 226)
Thank you for your suggestion. They are normally distributed, as you indicated, so Pearson's product-rate correlation was used, and corrected.
- Results
- Given the relative (in)accuracy and precision of measurement reporting values of blood glucose to 1dp is probably unwarranted. As data are presented as the mean (SD) the presumption is these data are normally distributed?
Answer: (line 226)
Thank you for your comment. They are normally distributed, as you indicated, so Pearson's product-rate correlation was used, and corrected.
- Figure 3 is the clearest representation of these data to the reader and provides the basis for consideration of the data outputs that follow. The count of occurrences per category of event in Table 3 is somewhat misleading as it does not consider the number of occurrences per event i.e., all occurrences of low blood glucose when competing (COMP) for a subject could have occurred during a single COMP session for that subject. This is compounded when these data are summated and then averaged over the number of participants whose data contribute to this analysis. Difficult to resolve a meaningful outcome to this level of analysis. Referring to Figure 3, the magnitude (not analysed) and time for which the blood glucose deviates from the ‘normal’ range appears minimal. On that basis have the authors considered the physiological relevance of the magnitude*time deviations from the ‘norm’ to the participant’s performance? (see discussion, which appear to concentrate on endurance performance, not CE and refers to ‘prolonged events of low blood glucose’)
Answer: (line 531)
Thank you for your valuable advice. This research focuses on blood glucose events that occur during intake, exercise, and sleep over a two-day (48-hour) competition. We believe that it is necessary to consider the intake strategy over a 48-hour period, rather than for each event, so we have provided a descriptive summary of the actions taken when a blood glucose event occurs.
In addition, in this study, the number of participants was not large enough to physiologically compare blood glucose levels and performance, and there is no previous research, so comparisons cannot be made. So, we believe it is meaningful to describe it descriptively. However, since it is necessary to physiologically evaluate the effect on performance in detail, we have added the need to repeat this test on the same athletes in different competitions as a future research topic.
Fourth, in this study, we were only able to measure 10 subjects during one competition, so it was difficult to evaluate in detail the physiological relationship between abnormal blood glucose levels and individual performance. In the future, it will be necessary to con-duct detailed analyzes based on individual performance levels by repeatedly conducting tests on the same athletes in different competition.
- The cumulative distribution analysis that follows represents (to this reader) further over-analysis of these observational data.
Answer: (figure 4)
Thank you for your advice. Based on previous research (doi:10.1177/1932296816648344), this chart was created because it was thought it would be a good way to visually show how far blood glucose levels deviate from the normal range on decathlon competition days. Based on this grouping, we would like to discuss the newly added average blood glucose levels and number of intakes.
- Figure 6 is important as these analyses forms the basis of the major outcome and conclusion (line 486/6), i.e., ‘Increased the frequency of energy-containing foods and beverages is essential to prevent the risk of low blood glucose during competition days.’ The evidence in support of this conclusion is weak. The average intake frequency of the 10 subjects are parsed into 3 groups of n=5, n=2 and n=3, respectively in relation to the % of events in which low (>10%) or high (>10%) or not classified blood glucose occurred, and subjected to statistical analysis by ANOVA (??)
Answer:
As the referee correctly pointed out, this does not lead to the main results and conclusions of this study. Therefore, we divided the incidence of blood glucose levels into two groups (under 80 mg/dL>10% vs. under 80 mg/dL≦10%) and compared the frequency of intake. We believe that by showing that athletes with 10% or more of under 80 mg/dL intake fewer meals (p<0.05), we will deepen the discussion about the rate of low blood glucose levels and the number of intake during competition. Accordingly, Fig. 4 (→Fig7: Pearson product-moment correlation coefficient) has been moved to 3.4 to show the relationship between blood glucose levels and meals. The following sentences have been added and changed to the results and discussion.
(line 344) One group consisted of five athletes of whom more than 10% of their measurements revealed low blood glucose levels (upper row: ID: 2, 5, 7, 8, and 9). Another group consisted of five athletes of whom less than 10% of their measurements revealed low blood glucose levels (under row: ID: 1, 3, 4, 6, and 10).
(line 350) Figure 5 shows that in these two groups, the average blood glucose level was higher in athletes whose blood glucose levels were under 80 mg/dL ≦10% (112.5 ± 8.4 mg/dL) than in those who were not (95.7 ± 5.7 mg/dL). (p<0.01).
Figure 6 shows the comparison of average intake frequency in these two groups. The number of intake for these two groups were higher for athletes with their blood glucose levels under 80 mg/dL ≦10% (15.0±5.0 mg/dL) than for those who did not (23.4±4.3 mg/dL) (p < 0.05).
Figure 7 shows the relationship between mean blood glucose level and the number of intakes. A significant positive correlation was observed between mean blood glucose levels and the number of intakes over the two days (r = 0.713, p < 0.021).
Do the authors consider these data are sufficient to support the conclusion offered?
Answer:
We apologize for the lack of supporting data. By adding analysis of changes in blood glucose levels and the number of intakes, we were able to further strengthen the conclusions of this study.
References:
- Unable to trace reference 11. Is this abstract/proceedings from a meeting?
Yale JF, Leiter LA, Marliss EB. Insulin resistance during recovery from strenuous exercise is greater in obesity. Intl Obes. 1985;9:A108 
Answer: (references)
Thank you for your suggestion. As you pointed out, this reference cannot be cited, so it was deleted. Instead, we added the mechanism by which blood glucose levels rise and described the possibility that high-intensity exercise can cause hyperglycemia.
(line 56) … high-intensity exercise causes rapid consumption of glucose in the blood and muscles, leading to a low blood glucose level, but it is thought that blood glucose level may rise temporarily as a result of the liver's response to maintain constant blood glucose levels.
16. Unable to trace reference 12 or the stated Journal. 12. Nagata K, Shiwa G, Otsuki C, Katsuhiko K. Classification of blood glucose levels in the Flash Glucose Monitoring(FGM)era―Focusing on hypoglycemia and blood glucose level spikes―. Comprehensive Medicine.2020;19(1):21-30
Answer: (references)
Thank you, it has been described in the answer to 3.

Reviewer 2 Report
Comments and Suggestions for Authors
Fluctuations in blood glucose levels during a decathlon competition; Focusing on the timing of intake and competing time
Issues:
- Although interesting, the study is a merely a descriptive one. Thus, this term “descriptive” should be included in the current title.
- Line 33. Should be “points scored” rather than “records”.
- Figure 2. Make it bigger so that reader can comprehend better.
- Line 174. “any” rather than “all”?
- Line 221. CV was not defined earlier.
Author Response
Dear Reviewer 2
We thank the referees for carefully reading our manuscript, now titled “Blood glucose levels during a decathlon competition: A descriptive study of timing for intake and competing time” and for their insightful comments. All corrections are highlighted in RED in the REVISED manuscript. Please check revised manuscuript.
Based on the referees’ comments, we have revised the manuscript and look forward to its future acceptance for publication in Metabolites.
Our responses to the referees’ comments are as follows:
Manuscript ID: metabolites-2763781
N Omi PhD
- Although interesting, the study is a merely a descriptive one. Thus, this term “descriptive” should be included in the current title.
Answer: (title)
Thank you for your suggestion. We have made the following revisions.
Fluctuations in blood glucose levels during a decathlon competition: A descriptive study of timing for intake and competing time.
- Line 33. Should be “points scored” rather than “records”.
Answer: (lines 33)
Thank you for your suggestion. We agree with your opinion. We have made the following revisions. points scored
- Figure 2. Make it bigger so that reader can comprehend better.
Answer: (Fig. 2)
Thank you for your suggestion. We increased the size of the figure.
- Line 174. “any” rather than “all”?
Answer: (line 186)
Thank you for your suggestion. We agree with your opinion. We have made the following revisions. any
- Line 221. CV was not defined earlier.
Answer: (line 165)
Thank you for your suggestion. We added the following text.
In addition, the coefficient of variation (CV; SD/Average blood glucose level × 100), which is the main blood glucose fluctuation index in the international consensus report, was calculated [14] (Table 2). Although there are individual differences for blood glucose levels, the subjects (n=10) analyzed in this study were within normal values (CV<36%).

Reviewer 3 Report
Comments and Suggestions for Authors
General comments
- English needs improvement. Consider having the article checked by a native speaker or use grammar application such as Grammarly or similar.
- Abstract
The abstract does not contain any background or hypothesis. Please add a sentence or two to cover these issues.
- Figure 1 (lines 146-147):
What is the purpose of putting the IT and a “rice ball” symbol in Figure 1? Although explained in the text, the figure is a little misleading since it seems that the intake was made only once on the day before the competition, and there were no intakes later. I suggest completely removing the “rice ball” from Figure 1.
- Figure 2: Inconsistent legend to the figure.
In the legend to Fig.2, there are “colors” for low (small number of black dots on a white background) and high (many black dots on a white background) glucose. However, no time interval on the figure is marked as low glucose. Additionally, the figure has grey regions not explained in the coloring legend. Please correct the inconsistency.
- Results
Lines 268-269: What is the purpose of pointing out the relationship between athletes’ height and blood glucose? Is there any correlation? Please provide references if this is the case.
- Results, section 3.4 (Line 345)
The number of participants was ten, which is relatively small. Yet, in the results, the authors divided these ten athletes into three groups, one of which comprised only two individuals. This number is too small for any serious statistical analysis. These data should be presented at the descriptive level.
- Discussion
Lines 415-416: Authors state that “low and high blood glucose levels are prevalent....”
For something to be declared as prevalent, it should be present more than 50% of the time, which is not the case in this study, as clearly shown in Figures 3 and 5. Please rephrase the statement.
- Discussion
Please discuss the importance and the influence of adrenaline and glucagon on gluconeogenesis in the liver and its production of glucose since it is known that “stress” increases blood glucose and that hepatocytes have receptors for both adrenaline and glucagon.
- Discussion
Individuals do not have identical metabolisms and operate at different blood glucose ranges. For example, a low glucose level for one person is not necessarily low for another. The same is true for high glucose.
- Discussion, Line 476
Authors discuss omitting the quantity of intake from the study. They should also comment on the composition of food intake.
Specific comments
- Line 33: There is an extra space between “scoring table, and the winners”.
- Line 266-267: Strange sentence.
- Line 335-6: missing word (suggestion underlined)
“Overall, high and low blood glucose levels were commonly observed at/during IT than competing time or sleep.”
- Line 342: Hard to understand sentence
“e People = Total number of people who intake.”
- Lines 355-357: Incomplete sentence.
“Figure 6 shows comparison of average intake frequency, conducted a between-subjects ANOVA on these conditions, revealing a significant difference among the (F(2,7)= 20.918, p < 0.001).”
- Lines 355-357: Incomplete sentence.
“Figure 6 shows comparison of average intake frequency, conducted a between-subjects ANOVA on these conditions, revealing a significant difference among the (F(2,7)= 20.918, p < 0.001).”
- Lines 425-427: Incomplete sentence.
“In this study, did not show any effect on performance, two athletes had low blood glucose within one hour after the start of the competing time within 45 minutes of intake, both with low blood glucose levels exceeding 10%.
- Lines 442-443: Incomplete sentence.
“This study would not be clear because daily energy intake was not assessed.”
- Line 446: word prevalent.
- Lines 484-485: Incomplete sentence.
Author Response
Dear Reviewer 3
We thank the referees for carefully reading our manuscript, now titled “Blood glucose levels during a decathlon competition: A descriptive study of timing for intake and competing time” and for their insightful comments. All corrections are highlighted in RED in the REVISED manuscript. Please check revised manuscuript.
Based on the referees’ comments, we have revised the manuscript and look forward to its future acceptance for publication in Metabolites.
Our responses to the referees’ comments are as follows:
Manuscript ID: metabolites-2763781
N Omi PhD

Round 2
Reviewer 1 Report
Comments and Suggestions for Authors
The authors have reacted positively to the initial review of this m/s. However, the intention of the commentary on the original manuscript was to invoke a reconsideration of the aims and objectives of the study and, specifically, to re-appraise the strength of the conclusion(s) based on weak, observational, data. The authors’ response and revised m/s considers each point of the review to be prescriptive. This was necessary to correct true error (of which there were many), but the authors could have made a greater effort in adopting the reviewer’s commentary to progress the argument further to improve, rather than just to correct, deficiencies in the m/s.
As indicate in the 1st paragraph of the initial review, the data presented is ‘descriptive’ in nature, i.e., there is no hypothesis to test, but the study itself is an ‘observation’ with no intervention/condition imposed by the observation of the blood glucose response of the decathletes during competition. I would recommend the title reflect this.
Examples where the revisions to the m/s could not be considered to have significantly improved the argument are presented below.
1. what is the value of the additional qualification and incorrect data value (i.e., how does 23 MET equate to 375.4 km/h?) presented in lines 40 through 43? Ref 1 is cited to confirm the physiological challenge of the decathlon (lines 35 through 39) to be of high intensity and subject to high intensity exercise fluctuations in glucose levels as stated in lines 56 through 69 and links to the new statement in line 315-6 ‘Overall, high and low blood glucose levels were commonly observed during IT rather than in the competing time or during sleep’ and further discussion of the relevance of this summary observation.
2. Lines 170 to 177 correct the previously misquoted physiological time delay between arterial and interstitial glucose, but does this matter in the overall context of these 2-day long observations with fixed-time epochs? Did this ‘correction’ have an impact on the outcome of the number of incidents of high/low glucose levels at any ‘phase’ of competition?
3. The authors have failed to address the concerns raised in the presentation of the number of occurrences of blood glucose events per subject per mode of activity. In its present form Table 3 and selective narrative of these data (3.3.) appear to have little to offer. In the rebuttal, the statements made (reproduced below) are unrelated to the issue this reviewer has raised, so this remains unresolved.
Figure 3 is the clearest representation of these data to the reader and provides the basis for consideration of the data outputs that follow. The count of occurrences per category of event in Table 3 is somewhat misleading as it does not consider the number of occurrences per event i.e., all occurrences of low blood glucose when competing (COMP) for a subject could have occurred during a single COMP session for that subject. This is compounded when these data are summated and then averaged over the number of participants whose data contribute to this analysis. Difficult to resolve a meaningful outcome to this level of analysis. Referring to Figure 3, the magnitude (not analysed) and time for which the blood glucose deviates from the ‘normal’ range appears minimal. On that basis have the authors considered the physiological relevance of the magnitude*time deviations from the ‘norm’ to the participant’s performance? (see discussion, which appear to concentrate on endurance performance, not CE and refers to ‘prolonged events of low blood glucose’)
Answer: (line 531)
Thank you for your valuable advice. This research focuses on blood glucose events that occur during intake, exercise, and sleep over a two-day (48-hour) competition. We believe that it is necessary to consider the intake strategy over a 48-hour period, rather than for each event, so we have provided a descriptive summary of the actions taken when a blood glucose event occurs.
In addition, in this study, the number of participants was not large enough to physiologically compare blood glucose levels and performance, and there is no previous research, so comparisons cannot be made. So, we believe it is meaningful to describe it descriptively. However, since it is necessary to physiologically evaluate the effect on performance in detail, we have added the need to repeat this test on the same athletes in different competitions as a future research topic.
Fourth, in this study, we were only able to measure 10 subjects during one competition, so it was difficult to evaluate in detail the physiological relationship between abnormal blood glucose levels and individual performance. In the future, it will be necessary to con-duct detailed analyzes based on individual performance levels by repeatedly conducting tests on the same athletes in different competition.
Modification to analysis represented by Figure 4 and narrative 3.4 promulgates the inaccuracy and over-analysis of these data. The principal outcome (lines 348-9), i.e., ‘Over half of the athletes spent more than 10% of their time with blood glucose levels outside the normal range of 80–139 mg/dL during the competition days (48 h)’ is not reported accurately. Analysis of Table 3 concludes,
i. the total records for the 10 subjects for which BG <= 80 mg/dL is 183, = average of 18.3 of total 192 (24*2*4) records per subject = 10%!
ii. The number of subjects for which BG <= 80 mg/dL is > 10% of their total records (i.e., > 19, is 5, i.e., 50% and range from 25 to 53, i.e., 12 to 28%).
iii. The remaining 5 subjects average 1% of total records for which BG <= 80 mg/dL, which is within the error of determination.
Based on analysis of frequency of records for which BG <= 80 mg/dL analyses the subjects are divided into 2 groups. What is the justification for group selection by this criterion? The depiction of the above parsed into these 2 sub-groups and as cumulative distribution plots (Figure 4) is further example of over-analysis to no effect.
Note: The description of grouping ‘low’ or ‘high’ blood glucose is misleading. The criterion is as stated in line 343-4 ‘The athletes were divided into two groups based on their blood glucose (under 80 mg/dL) levels.’
The subsequent analysis (Figure 5 and 6) of ‘these two groups’ (line 350 and 353) compounds the above as it (arbitrarily) declares this criterion relates to the overall blood glucose response and average intake frequency, respectively, for these subjects over the 48h period of observation.
Note: Please state what the error bars represent within the legend to Figures.
Figure 7 and analysis appears relevant, but it should be noted that the association between these two variables, though statistically significant, has an R2 of 0.5. There are other factors involved!
The discussion has been extended to include an inference to glucoregulation and diabetes risk that is probably beyond the scope of this observational study and its outcomes.
Comments on the Quality of English LanguageIt would be advisable to seek a review of the m/s to ensure basic grammar and, more importantly, that the arguments present within incorrectly structured sentences are not misinterpreted by the reader.
Author Response
Dear Reviewer 1:
We thank the referee for a second critical reading of our manuscript. We have carefully read your comments and have made revisions in the attached manuscript (metabolites-2763781) with a ‘re-phrased TITLE’ as per your advice to reflect the ‘nature’ of this present study: “Blood glucose levels during decathlon competition: An observational study of timing for intake and competing time”.
All corrections (including some minor errors we found during a re-reading of the text) are marked in RED in the manuscript. We hope that our manuscript will be considered for publication in Metabolites. Thank you very much.
- As indicate in the 1st paragraph of the initial review, the data presented is ‘descriptive’ in nature, i.e., there is no hypothesis to test, but the study itself is an ‘observation’ with no intervention/condition imposed by the observation of the blood glucose response of the decathletes during competition. I would recommend the title reflect this.
Answer:
Thank you for your suggestion, and we understand your point of view that our manuscript is indeed an ‘observational’-type study. Therefore, the title has been corrected as follows:
“Blood glucose levels during decathlon competition: An observational study of timing for intake and competing time”
- what is the value of the additional qualification and incorrect data value (i.e., how does 23 MET equate to 375.4 km/h?) presented in lines 40 through 43? Ref 1 is cited to confirm the physiological challenge of the decathlon (lines 35 through 39) to be of high intensity and subject to high intensity exercise fluctuations in glucose levels as stated in lines 56 through 69 and links to the new statement in line 315-6 ‘Overall, high and low blood glucose levels were commonly observed during IT rather than in the competing time or during sleep’ and further discussion of the relevance of this summary observation.
Answer: (lines 39)
Thank you for your correction. The 375.4 km/h added last time in revision 1, was in error. The correct value is 22.5 km/h (375.4 m/min). We deeply apologize for the error in the description. It is also noted in the METs Table [4] that 23 METs is equivalent to 22.5 km/hour, which was the highest intensity of all the competitive events listed in the METs Table. Since the subjects in this study were running at roughly 28 km/hour (400 m sprints), we used this as one of the guidelines for high intensity. Previously, we were asked to indicate why the mixed events were high intensity, but since there was no previous research to clearly indicate this, we used METs to indicate this.
- Lines 170 to 177 correct the previously misquoted physiological time delay between arterial and interstitial glucose, but does this matter in the overall context of these 2-day long observations with fixed-time epochs? Did this ‘correction’ have an impact on the outcome of the number of incidents of high/low glucose levels at any ‘phase’ of competition?
Answer: (lines 168)
Thank you once again. As you pointed out, it is not necessary in the overall context observed over 2 days, since it does not affect the number of hypoglycemia・hyperglycemia occurrences. Therefore, “In recent years, various studies have shown that long-term measurements on subjects such as diabetic patients and children can minimize measurement errors [15]” has been deleted.
- The authors have failed to address the concerns raised in the presentation of the number of occurrences of blood glucose events per subject per mode of activity. In its present form Table 3 and selective narrative of these data (3.3.) appear to have little to offer. In the rebuttal, the statements made (reproduced below) are unrelated to the issue this reviewer has raised, so this remains unresolved.
Modification to analysis represented by Figure 4 and narrative 3.4 promulgates the inaccuracy and over-analysis of these data. The principal outcome (lines 348-9), i.e., ‘Over half of the athletes spent more than 10% of their time with blood glucose levels outside the normal range of 80–139 mg/dL during the competition days (48 h)’ is not reported accurately. Analysis of Table 3 concludes,
- the total records for the 10 subjects for which BG <= 80 mg/dL is 183, = average of 18.3 of total 192 (24*2*4) records per subject = 10%!
- The number of subjects for which BG <= 80 mg/dL is > 10% of their total records (i.e., > 19, is 5, i.e., 50% and range from 25 to 53, i.e., 12 to 28%).
iii. The remaining 5 subjects average 1% of total records for which BG <= 80 mg/dL, which is within the error of determination.
Based on analysis of frequency of records for which BG <= 80 mg/dL analyses the subjects are divided into 2 groups. What is the justification for group selection by this criterion? The depiction of the above parsed into these 2 sub-groups and as cumulative distribution plots (Figure 4) is further example of over-analysis to no effect.
Answer:
Yes, the Referee is right in pointing out our over-analysis, and the comment is deeply appreciated. Table 3 is a count of the number of times below 80 mg/dL and above 139 mg/dL that was observed in the individual 192 epoch readings of the 10 athletes used in this study, categorized by what they were doing at the time. As you pointed out, it is inappropriate to discuss the total number of times or the average value, and we have revised Table 3 to show only the number of times observed and the number of people.
And, we have also modified the cumulative distribution plot in Figure 4 to visually represent the characteristics of the 48 h blood glucose levels, instead of dividing them by the percentage of occurrence of hypoglycemia. Some of the wording of the discussion have been changed to be appropriate for an observational study.
- Note: The description of grouping ‘low’ or ‘high’ blood glucose is misleading. The criterion is as stated in line 343-4 ‘The athletes were divided into two groups based on their blood glucose (under 80 mg/dL) levels.’
The subsequent analysis (Figure 5 and 6) of ‘these two groups’ (line 350 and 353) compounds the above as it (arbitrarily) declares this criterion relates to the overall blood glucose response and average intake frequency, respectively, for these subjects over the 48h period of observation.
Answer: (Figure 5 and 6)
Thank you for pointing this out. The wording you pointed out was highly misleading and inappropriate, so we have deleted it. In addition, we deleted the analysis comparing two groups based on the percentage of occurrence of hypoglycemia, as there was no clarity in the reference value, and discussed by looking at the association.
- Note: Please state what the error bars represent within the legend to Figures.
Answer: (Figure 5 and 6)
We apologize for the omission. We have removed this figure as it is an arbitrary analysis as you pointed out.
- Figure 7 and analysis appears relevant, but it should be noted that the association between these two variables, though statistically significant, has an R2 of 0.5. There are other factors involved!
Answer: (lines 437)
Thank you for your advice. We evaluated only the correlation coefficient and failed to consider the coefficient of determination. Therefore, we have rephrased the text of the discussion to:
Although other factors may be involved, highlighting the number of intakes on competition days in the decathlon, a higher number of intakes correlated with higher average blood glucose levels. Therefore, it was considered important to ensure the frequency of intake, even during the short rest periods between competitions.
- The discussion has been extended to include an inference to glucoregulation and diabetes risk that is probably beyond the scope of this observational study and its outcomes.
Answer: (lines 432)
Other reviewers pointed out that we should include this discussion, and we did so after reviewing the process of blood glucose levels. However, as the Referee stated, it is not appropriate for the intent of an observational study on decathletes and is not what the authors wanted to discuss, so we have deleted the unnecessary text and re-phrased therein in the 4th paragraph of the Discussion section.
Additionally, it can be assumed that both the action of glucagon [25,26] and the action of adrenaline associated with sympathetic activation [27], such as tension, increased the blood glucose levels. Although we did not take real-time measurements as the competition was in progress, it can be inferred that both action of glucagon and adrenaline influenced blood glucose levels.
